# A qualitative study on the perspectives of prenatal breastfeeding educational classes in Ireland: Implications for maternal breastfeeding decisions

Jennifer Kehinde[1], Claire O'Donnell[1], Annmarie Grealish[1,2]*

1 Department of Nursing and Midwifery, Health Research Institute, University of Limerick, Limerick, Ireland,
2 Kings Florence Nightingale Faculty of Nursing, Midwifery & Palliative Care, King's College London, London, United Kingdom

* annmarie.grealish@ul.ie

**Data Availability Statement:** All relevant data are within the manuscript and its Supporting Information files.

## Abstract

### Background

Breastfeeding, acknowledged for its critical health benefits for both infants and mothers, remains markedly underutilized in Ireland, which reports the lowest breastfeeding rates in Europe. Recent data indicate that fewer than 60% of Irish mothers initiate breastfeeding at birth, with this rate precipitously declining in the subsequent weeks postpartum. Various sociocultural, psychological, and educational elements, such as prenatal breastfeeding education, influence this persistently low prevalence. This descriptive qualitative study explores the perspectives of mothers and healthcare professionals, specifically midwives and lactation consultants, on prenatal breastfeeding education classes in Ireland and how they influence mothers' breastfeeding decisions.

### Methods

A qualitative descriptive methodology was employed, utilizing online semi-structured interviews with midwives, lactation consultants (n = 10), and postnatal mothers (n = 20) from four tertiary hospitals in the Republic of Ireland. The data were subjected to reflexive thematic analysis, adhering to the six-step process of thematic analysis, to extrapolate and analyse the interview transcriptions. NVivo software was used to facilitate this analysis, given its robust capabilities in organizing, coding, and retrieving qualitative data efficiently. Four criteria for qualitative research were also used to enhance analytical rigor.

### Results

Prenatal breastfeeding education in Ireland often presents breastfeeding in an idealized way, resulting in a gap between mothers' expectations and their actual experiences. Participants needed practical content that included realistic scenarios and breastfeeding benefits. Additionally, findings indicate a desire for a more interactive and personalized educational model to address expectant mothers' unique needs better. Limitations of the virtual class

**Funding:** The author(s) received no specific
funding for this work.

**Competing interests:** The authors have declared
that no competing interests exist.

format were also highlighted, with participants noting the potential for technology to improve
engagement and personalization. The data further underscore the need for consistency and
accuracy in breastfeeding education, with participants identifying standardized approaches
and awareness of socio-cultural dynamics, including partner involvement, as essential ele-
ments in effective prenatal breastfeeding education.

## Conclusion

This study underscores the necessity for a more realistic, interactive, and standardized
approach to prenatal breastfeeding education in Ireland. Current classes often set idealized
expectations that may leave mothers feeling unprepared for breastfeeding's real-life chal-
lenges. Addressing these gaps by integrating practical scenarios, enhancing technological
tools for virtual classes, and incorporating socio-cultural considerations could improve
breastfeeding education and maternal outcomes. This qualitative descriptive study high-
lights a disconnect between educational objectives and mothers' actual experiences, advo-
cating for a holistic approach that includes personalized, culturally sensitive support and
comprehensive training for healthcare providers.

## Introduction

Extensive scientific research has consistently demonstrated the unparalleled benefits of breast-
feeding as the optimal source of infant nutrition [1–3]. This compelling evidence establishes a
strong connection between breastfeeding and promoting optimal infant growth and develop-
ment, significantly contributing to improved maternal health outcomes [4, 5]. The World
Health Organization (WHO) recommends exclusive breastfeeding for the first six months of
life, followed by continued breastfeeding and appropriate complementary foods up to two
years of age or beyond [6, 7]. These recommendations are backed by robust empirical evi-
dence, demonstrating that breastfeeding reduces the risk of infections and chronic diseases [7,
8], enhances cognitive development in children [9, 10], and lowers the risk of breast and ovar-
ian cancers in mothers [7, 11, 12].

Despite the acknowledged advantages of breastfeeding, overall breastfeeding initiation and
continuation rates in many developed countries, including Ireland, continue to fall short of
optimal levels [13–15]. According to a 2023 World Breastfeeding Trends Initiative (WBTI)
assessment, only approximately 40% of infants in Ireland are breastfed at three months,
whereas a substantial 60% are formula-fed [16]. These figures highlight a strong cultural pref-
erence for formula feeding within Irish society. In collaboration with Technological University
Dublin and UNICEF, the World Breastfeeding Trends Initiative published its inaugural report,
ranking Ireland 57th out of 99 participating countries worldwide, with a 56 out of 100 score
for breastfeeding support and protection [17]. While these findings highlight the urgent need
to enhance breastfeeding support and promotion in Ireland, they also present a clear opportu-
nity for improvement. Furthermore, Ireland is ranked 10th among the 19 European nations
included in the evaluation [16]. Similarly, Ireland consistently records the lowest breastfeeding
initiation rate among European countries, at just 60% [16]. This rate is markedly lower than
the breastfeeding initiation rates documented in Australia, the United Kingdom, and the
United States [18, 19]. According to the Australian Bureau of Statistics, in 2022, 90.6% of chil-
dren aged 0–3 years had received breast milk at some point [20]. In the United Kingdom, the

latest data reveals that the breastfeeding initiation rate is 72% from 2019/20 to 2023/24 [21]. This reflects the proportion of infants who received their first feed from maternal or donor breast milk immediately after birth. In the United States, the Centres for Disease Control and Prevention (CDC) reported that 83.2% of infants born in 2019 were breastfed at some stage. However, exclusive breastfeeding rates significantly decline as infants age [22]. According to the findings presented in the Irish Maternity Indicator System report in 2020, a considerable proportion of mothers, precisely 60%, initiated breastfeeding postpartum. The rates plummeted to 43% at three months postpartum, further declining to 6% of newborns [23].

The choice to initiate and maintain breastfeeding is not simple; rather, it is a decision influenced by a complex interplay of factors, including sociocultural, psychological, and educational elements such as prenatal breastfeeding education [8, 24, 25]. Prenatal breastfeeding education has emerged as a pivotal determinant of breastfeeding outcomes [24, 25]. Previous research has identified several factors that can influence the success of breastfeeding education, including the content and delivery of the classes, the timing of the education, and the extent to which they address the mother's educational needs [26–28]. Studies have shown that mothers attending prenatal breastfeeding education classes are more likely to initiate breastfeeding and continue it longer than those who do not receive such education [24, 25, 29]. The effectiveness of the prenatal breastfeeding program on exclusive breastfeeding rates was investigated to assess the patterns and incidences of breastfeeding among the participants (n = 60) [30]. The study revealed a marked improvement in the uptake of exclusive breastfeeding in the experimental group (n = 30, 83.3%) compared to the control group (n = 30, 36.7%) following birth. The experimental group exhibited a higher incidence of exclusive breastfeeding than the control group. Significantly, 90% of infants in the experimental group completed their first feed within two hours postpartum. At discharge, 93.3% of mothers in the experimental group and 53.3% in the control group were exclusively breastfeeding. One month postpartum, 83.3% of mothers in the experimental group continued exclusive breastfeeding, compared to 36.7% in the control group.

Healthcare professionals are critical in providing educational support and addressing specific breastfeeding challenges to sustain breastfeeding [8, 24, 31–33]. Studies [3, 24, 31, 34, 35] emphasize the essential role of knowledgeable and skilled healthcare professionals in optimizing breastfeeding outcomes and are integral in the early detection and management of prevalent breastfeeding challenges, including latching difficulties, nipple pain, and perceived insufficient milk supply. Studies [36–38] highlight the significant role of continuous professional support in enhancing breastfeeding duration and exclusivity and the criticality of this support in the early postpartum period, where timely interventions can make a substantial difference in breastfeeding success.

Prenatal breastfeeding education in developed countries incorporates various methods tailored to different learning preferences and needs. Traditional in-person classes, facilitated by healthcare professionals such as lactation consultants, offer comprehensive information and practical demonstrations [39, 40]. Online courses and webinars have gained considerable traction due to their flexibility, providing videos, interactive modules, and virtual Q&A sessions [41, 42]. Additionally, mobile applications are innovative, offering easily accessible information and support through instructional videos, articles, and real-time chat with lactation consultants, thus enhancing user engagement and support [43]. Furthermore, printed materials, including brochures and pamphlets, continue to serve as valuable supplements to these educational methods [44–46]. Various prenatal breastfeeding education resources are available in Ireland, including group classes, online classes, and individual consultations with lactation consultants; however, their effectiveness has not yet been evaluated [46, 47]. This current study is the first to examine midwives'/lactation consultants' and mothers' perspectives on prenatal breastfeeding educational classes in Ireland.

Implementing various breastfeeding initiatives evidences Ireland's commitment to fostering a supportive breastfeeding environment. The most recent being the formulation of the Breastfeeding Action Plan and the National Maternity Strategy [48, 49]. These initiatives in Ireland aim to achieve an annual 2% increase in breastfeeding rates. However, our recent systematic review [24] concluded that no published evaluation of the impact of these most recent strategies on breastfeeding rates has been published. Although several studies [50–56] explored various aspects of breastfeeding in Ireland, none investigated prenatal breastfeeding educational classes. This current study examined the current prenatal breastfeeding education classes in Ireland, placing significant importance on the perspectives of midwives and postnatal mothers about the mother's breastfeeding decisions. This study explored midwives'/lactation consultants' and postnatal mothers' perspectives on prenatal breastfeeding educational classes in Ireland, particularly how these classes are perceived to inform or support mothers' breastfeeding decisions.

## Methods

### Study design

A descriptive qualitative methodology study design using Braun and Clarke's reflexive thematic analysis [57] was used to capture mothers' and lactation consultants'/midwives' perspectives of the prenatal breastfeeding education classes in Ireland and their impact on mothers' decision to breastfeed. This study is reported in accordance with the consolidated criteria for reporting qualitative research (COREQ) [58] using the 32-item checklist to enhance the transparency of our study's reporting (S1 Table). This study, rooted in the interpretivist paradigm, sought to understand phenomena through the meanings people assign to them [59, 60]. The interpretivist paradigm centred on understanding individuals' perspectives, beliefs, and values is particularly relevant to this qualitative study. The study focuses explicitly on lactation consultants, midwives, and mothers, making the interpretivist paradigm fitting. This approach facilitates an in-depth exploration of individual viewpoints, enhancing understanding of subjective perspectives on prenatal breastfeeding education classes and the perceived role of these classes in shaping breastfeeding decisions. The study acknowledges the complex interplay of personal, social, environmental, and cultural factors influencing mothers' breastfeeding decisions.

### Study setting and selection of participants

Online semi-structured qualitative interviews were conducted with lactation consultants/midwives who directly delivered prenatal breastfeeding and postnatal mothers (0–6 months) who attended the prenatal breastfeeding education classes. Four tertiary hospitals in the Republic of Ireland (University Maternity Hospital Limerick, Rotunda Maternity Hospital Dublin, University Hospital Galway, and Coombe Women and Infant University Hospital Dublin) were selected for this study for sociocultural diversity and a unique mix of urban and rural communities to ensure a comprehensive representation of diverse perspectives. The eligibility criteria used purposive and snowball sampling [61] to purposefully select each participant, as illustrated in Table 1, as it enabled the researchers to intentionally choose individuals who possess pertinent experiences and narrative competence about the research topic.

### Recruitment

Recruitment took place from October 2022 to July 2023. The principal investigator (PI) (*Blinded*) provided all potential participants (postnatal mothers and healthcare professionals) with the Participant Information Leaflet (PIL), developed in collaboration with the patient and participant involvement (PPI) representative. The PIL contained the study's aim/

**Table 1. Eligibility criteria for participants.**

| Types of participants | Inclusion criteria | Exclusion criteria |
|---|---|---|
| **Healthcare Professionals** | • Lactation Consultants and Midwives currently employed at the selected research sites.<br>• Lactation Consultants and Midwives actively involved in providing prenatal breastfeeding education to mothers at the selected research sites.<br>• Provision of informed consent. | • Lactation Consultants and Midwives not currently employed at the selected research sites.<br>• Lactation Consultants and Midwives not actively involved in providing prenatal breastfeeding education to mothers at the selected research sites.<br>• Declined or did not provide informed consent |
| **Postnatal Mothers (0–6 months postpartum).** | • Postnatal mothers, including multigravidas (women who have been pregnant more than once) and primigravidas (women who are experiencing their first pregnancy), who attended prenatal breastfeeding education classes at the selected research sites.<br>• Provision of informed consent | • Postnatal mothers who did not attend prenatal breastfeeding education classes at the selected research sites.<br>• Declined or did not provide informed consent. |

objectives, participant criteria, and PI's (*Blinded*) contact details. The directors of midwifery gave the PI direct access to all healthcare professionals (Lactation consultants/midwives) and postnatal mothers in the postnatal wards. Ten lactation consultants/midwives (n = 10) and twenty postnatal mothers (n = 20) agreed to participate and met the eligibility criteria. The PI obtained written informed consent (*Blinded*) before each interview, and verbal consent was recorded at the beginning of each interview.

## Patient and Public Involvement (PPI)

Two PPI advisors, including two postnatal mothers, one lactation consultant, and one midwife experienced with prenatal breastfeeding education classes, were set up to amplify the voices of public members and improve the study's quality and relevance to participants [62] and met four times during the duration of the study. The PPI group were asked to give advice, provide feedback on the interview topic guide, and collaborate on the recruitment process. Details of the PPI group and their contributions and impact on the research were reported using the Guidance for Reporting Involvement of Patients and the Public (GRIPP2) (S2 Table).

## Data collection

The PI conducted all individual semi-structured interviews with all participants online using Microsoft Teams between July 2023 and November 2023. The PI used interview topic guides (S3 Table), which was informed by our recent systematic review [24] and further piloted with the two PPI advisors, resulting in minor adjustments to the language and the order of questions. The PI audio-recorded all semi-structured interviews. All participants' demographic information was obtained through an online questionnaire via Qualtrics before the interview.

Additionally, postnatal mother participants were asked to complete the Iowa Infant Feeding Attitude Scale [63] via Qualtrics before interviews to validate further and strengthen the qualitative data (S4 Table). The Iowa Infant Feeding Attitude Scale (IIFAS) measures attitudes toward infant feeding methods and breastfeeding versus formula feeding. The IIFAS consists of 17 statements rated on a 5-point Likert scale, ranging from 1 (strongly disagree) to 5 (strongly agree). Healthcare professional participants were asked to complete the adapted version of the HSE's Self-Assessment Competency Framework, obtained via Qualtrics before the interview (S5 Table). This scale helped to assess the healthcare professional's level of

competence in breastfeeding education and support skills by asking them to rate their agreement with each of the ten competencies statements reflecting their competence on a scale from "Strongly Disagree" to "Strongly Agree".

None of the participants expressed distress during the data collection process. Participants were offered the opportunity to review their transcripts; only two participants accepted the offer and expressed satisfaction with the content and accuracy of their transcripts upon review. Throughout this process, the PI maintained a reflexive journal to facilitate self-awareness, methodological transparency, data interpretation, and trustworthiness, ultimately enhancing the rigor and credibility of the research findings.

### Data analysis

All transcribed interviews were anonymized for data management purposes, transferred to NVivo (Version 14; QSR, 2024), and stored in a password-protected database. The transcripts were coded with numerical identifiers corresponding to the respective digital audio recordings to maintain the participants' anonymity. Thematic saturation was achieved through an iterative, inductive approach. Data were analysed continuously after each interview, allowing themes to be constructed and refined based on participant responses. Reflexive thematic analysis supported deep engagement with the data, enabling the development of rich, nuanced themes that captured participants' diverse perspectives. Saturation was reached when additional interviews added no new, substantial insights to the existing themes, indicating that each theme was thoroughly represented and grounded in the data [57, 64].

The data analysis used reflexive thematic analysis and the six-step process of thematic analysis to extrapolate and analyse the data from the interview transcriptions [57]. The PI conducted the initial analysis. Subsequent codes and themes were deliberated and refined by XX and XX (*BLINDED*), all proficient in qualitative data analysis, and conducted the final review of the themes. Data analysis (Fig 1) began with (1) initial coding after reading and data familiarization, (2) data coded for patterns on how BF classes are delivered and experienced, and the requirements needed to increase/ enhance the uptake of BF in perinatal clinical routine care, (3) the codes were refined and used to construct initial themes and to answer the research question, (4) we then inductively analysed the data for patterns and consistency within the entire dataset, (5) naming and defining the themes, (6) the results were refined and results presented with excerpts of verbatim text to display thematic meaning.

To enhance analytical rigor, Lincoln and Guba's [65] four criteria of qualitative research (credibility, dependability, confirmability, and transferability) were also used (Table 2).

### Ethical approval

Ethical approval was obtained from the Research Ethics Committees of four tertiary hospital sites in the Republic of Ireland namely, HSE Research Ethics Committee University Maternity Hospital Limerick (Reference Number: Ref-119/2021), HSE Research Ethics Committee University Hospital Galway (Reference Number: C.A. 2718), Rotunda Maternity Hospital Research Ethics Committee (Reference Number: Ref-2021-027) and Research Ethics Committee Coombe Women and Infant University Hospital (Reference Number: Study No.18-2021). All procedures followed the Declaration of Helsinki regulations [66].

### Results

Table 3 provides details of all 20 postnatal mothers and 10 healthcare professionals (lactation consultants/ midwives) who participated in the study from four tertiary hospitals (n = 30).

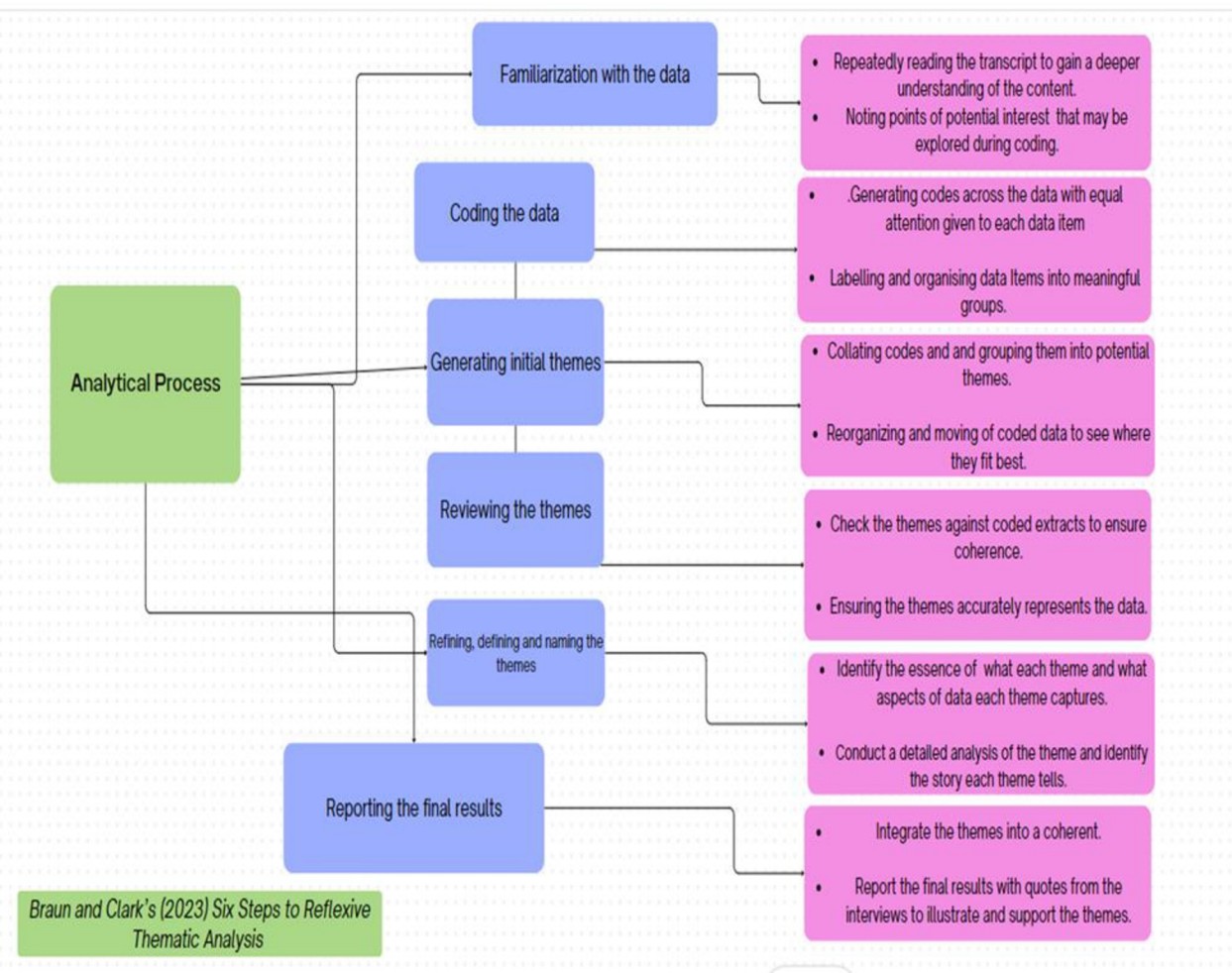

**Fig 1. Data analysis process.**

## Participant characteristics (Healthcare *professionals* and postnatal mothers)

All participants' (n = 30) demographic characteristics are presented in Table 3. All ten healthcare professionals were female (n = 10) and worked in the maternity unit across all selected research sites. They were directly involved in providing prenatal breastfeeding education to mothers. Six are double qualified (midwives/lactation consultants); three are lactation consultants, and one is a midwife. The healthcare participants were aged between 31 and 52 years (mean age, 40.1), and the majority of the participants were Irish (n = 7, 70%), any other white background (n = 2; 20%) and Asian (Indian) (n = 1; 10). All lactation consultants and midwives were affiliated with the Nursing and Midwifery Board of Ireland (NMBI) or the Association of Lactation Consultants in Ireland. The educational attainment of lactation consultants and midwives ranged from a Master's degree (n = 6; 60%) and a Bachelor's degree (n = 4; 40%). Years working as a lactation consultant/ midwife ranged from 4 to 8 years (Mean years, 5.56). Ten (n = 10) participants (healthcare professionals) completed the adapted version of the HSE's self-assessment competency framework [67] for breastfeeding support before being interviewed. For each of the ten competencies, participants indicated their level of agreement

**Table 2. Analytical rigor.**

| Quality Criteria | Action taken by the researchers |
|---|---|
| Credibility | • Including HCPs who are directly involved in providing prenatal breastfeeding education.<br>• Including postnatal mothers who actively engaged with and attended the prenatal breastfeeding education classes before delivery.<br>• **Triangulation:** comparing data from interviews with postnatal mothers and healthcare professionals,<br>• **Member Checking:** Sharing interview transcripts or summaries of the findings with the participants to confirm that their perspectives have been accurately captured and interpreted.<br>• **Peer Debriefing:** Regularly consultation with the research team to review and discuss the data and emerging themes.<br>• Use of quotes to support the findings and ensure data accuracy.<br>• **Prolonged Engagement**: Implementing a sustained engagement strategy with the participants. |
| Dependability | • Detailed description of the study methodology and Philosophical underpinnings.• Application of reflexive thematic analysis discussed in the data analysis section.• The PI kept a reflexive diary and took field notes during and after interviews, including notes on theme identification, what was influencing them, and how this impacted the data [57].<br>• **Audit Trail:** Keep detailed records of all aspects of your research process, including raw data (e.g., interview recordings and transcripts), field notes, coding schemes, and decisions made during data analysis.<br>• **Code-Recode Strategy**: Compared new codes with the initial codes to identify any discrepancies or changes in interpretation. This process ensured the stability and reliability of your coding framework. |
| Confirmability | • All authors read the transcripts independently, ensuring the codes' appropriateness.<br>• Thematic analysis was applied as described in the data analysis.<br>• Codes, themes and subthemes were discussed among the researchers to avoid bias in interpretation and inter-coder agreement was used to confirm the final themes and subthemes. |
| Transferability | • Verbatim transcriptions were used to ensure the accuracy of the data.<br>• Descriptions of the research setting in which the participants were recruited.<br>• The characteristics of the study participants were provided. |
| Reflexivity | • As proposed by Braun and Clark [57], reflexivity was used to increase the quality of the qualitative evidence.<br>• The interpretation was questioned to ensure data accuracy and avoid interpretation bias.<br>• The lead author maintained a reflective diary. |

with statements reflecting their competence, ranging from 'Strongly Disagree' to 'Strongly Agree' (S5 Table). All ten competency areas yielded unanimous results. In this regard, all midwives indicated 'Strongly agree' with each statement. The consistent responses by all lactation consultants/midwives reflect their collective confidence in their ability to provide breastfeeding education and counselling. A total of 510 minutes of interview content was collected from the healthcare participants, and the median interview length was 45 minutes (ranging from 20–60 minutes).

The postnatal mothers were aged between 24–42 (mean age, 32.6). Most of the postnatal mothers were married (n = 13; 65%) and cohabiting (n = 7; 35%) and identified as Irish (n = 12; 60%), Black (Irish, African) (n = 6; 30%) and Asian (Indian) (n = 2; 10%). The education status of postnatal mothers ranged from a degree level (n = 11; 55%), postgraduate degree (n = 4; 20%), second-level education (Junior/Leaving Certificate) (n = 4; 20%) to primary education (n = 1; 5%). Fourteen of the postnatal mothers identified as in full-time employment (n = 14; 70%), stay-at-home mothers (n = 3; 15%), unemployed (n = 2; 10%) and one mother (n = 1; 5%) did not disclose her employment status. Nine postnatal mothers (n = 9; 47.4%) identified as Primiparous (first-time mothers), while eleven mothers (n = 11; 52.6%) identified as Multiparous (Mothers with previous pregnancies). A total of 920 minutes of interview content was collected from the postnatal mothers (n = 20), and the median interview length was 40 minutes (ranging from 18–55 minutes).

**Table 3. Participant demographic information.**

| Study participants | HealthCare Professionals (Lactation Consultants and midwives) | Postnatal Mothers |
|---|---|---|
| **Number of Interview participants** | 10 | 20 |
| **Age (mean)** | 40.1 (5.2) | 34.8 (4.6) |
| **Gender (%)** | | |
| Female | 10(100%) | 20 (100%) |
| Male | - | - |
| **Years of Experience (%)** | | |
| 2–4 years | 5(50%) | - |
| 5–7 years | 3(30%) | - |
| 8–10 years | 2 (20%) | - |
| **Ethnicity (%)** | | |
| Irish | 7 (70%) | 12 (60%) |
| Black | - | 6 (30%) |
| Asian (Indian) | 1(10%) | 2 (10%) |
| Other White Background | 2 (20%) | - |
| **Marital Status n, %** | | |
| Married | - | 13 (65%) |
| Co-habiting | - | 7 (35%) |
| **Educational Attainment (%)** | | |
| Leaving Cert | - | 2 (10%) |
| Junior Cert | - | 2 (10%) |
| Primary School | - | 1 (5%) |
| Degree | 4 (40%) | 11 (55%) |
| Postgraduate Degree | - | 4 (20%) |
| MSc | 6(60%) | - |
| **Professional Role (%)** | | |
| • Registered Midwife on the Nursing and Midwifery Board of Ireland (NMBI)/Certified Lactation Consultants on International Board-Certified Lactation Consultant. | 6 (60%) | - |
| • Certified Lactation Consultants on International Board-Certified Lactation Consultant. | 3 (30%) | - |
| • Registered Midwife on the Nursing and Midwifery Board of Ireland (NMBI) | 1 (10%) | - |
| **Employment Status n, %** | | |
| Employed | - | 14 (70%) |
| Stay Home Mother | - | 3 (15%) |
| Unemployed | - | 2 (10%) |
| No Response | - | 1 (5%) |
| **Parity (Number of Pregnancies) n, %** | | |
| **Primiparous** (A woman who has given birth once). | - | 9 (45%) |
| **Multiparous (**A woman who has given birth two or more times**).** | - | 11(55%) |
| **Intention to Breastfeed (Prior Delivery) n, %** | | |
| Combined Feed (Breastfeed and Formula Feed) | - | 8 (40%) |
| Exclusively Breastfeed | - | 2 (10%) |
| Formula Feeding | - | 6 (30%) |
| Unsure/Undecided | - | 4 (20%) |
| **Infant Feeding Method (Post Delivery) n, %** | | |
| Combined Feed (Breastfeed and Formula Feed) | - | 11(55%) |

*(Continued)*

**Table 3.** (Continued)

| Study participants | HealthCare Professionals (Lactation Consultants and midwives) | Postnatal Mothers |
|---|---|---|
| Exclusively Breastfeed | - | 3 (15%) |
| Formula Feeding | - | 6 (30%) |
| **Breastfeeding In Public n, %** | | |
| No-never breastfed in a public place | - | 16 (80%) |
| Yes- breastfed in a public place | - | 2 (10%) |
| Yes- bottle-fed expressed breastmilk in a public place | - | 2 (10%) |
| **Public Breastfeeding Experience n, %** | | |
| ***Difficulty Finding a Suitable Place to Breastfeed in Public n, %*** | | |
| Yes | - | 2 (10%) |
| No | - | 1 (5%) |
| I have Never Tried to Breastfeed in a Public Place | - | 17 (85%) |
| ***Experience of Being Stopped or Made Uncomfortable While Breastfeeding in Public n, %*** | | |
| Yes, I have | - | 1 (5%) |
| No, I Have Not | - | 2 (10%) |
| I Have Never Tried to Breastfeed in a Public Place | - | 17 (85%) |
| ***Factors Discouraging Breastfeeding in Public n, %*** | | |
| I Have Never Tried to Feed My Baby in a Public Place | - | 17 (85%) |
| Lack of Suitable Place Available to Breastfeed | | No Data |
| Being Made Uncomfortable by Other People | - | 2 (10%) |
| Being Stopped or Asked Not to Breastfeed | - | No Data |
| Concerns About Hygiene in Public Places | - | No Data |
| Other Reasons | - | No Data |
| **The Iowa Infant Feeding Attitude Scale (IIFAS)** | - | Mean score 62 (SD 8.0) |
| **The HSE's Self-Assessment Competency Framework.** | 10 (100%) | - |

Influenced by various considerations and personal preferences, the postnatal mothers reported different infant feeding intentions during pregnancy. Eight of the postnatal mothers (n = 8; 40%) indicated the intention to use a combination of breast milk and formula feeding, two (n = 2; 10%) planned to breastfeed exclusively, six (n = 6;30%) intended to solely formula feed, and four (n = 4; 20%) were uncertain or undecided about their infant feeding intention. Regarding the postnatal mother's infant feeding method after delivery, eleven (n = 11; 55%) reported using a combination of breastmilk and formula, three (n = 3; 15%) reported exclusively breastfeeding and six of the participants (n = 6; 30%) reporting formula feeding. Sixteen of the postnatal mothers reported that they had never breastfed in a public place (n = 16; 80%), two reported they had breastfed in a public setting (n = 2; 10%), two (n = 2; 10%) indicated that they had bottle-fed expressed breastmilk in public spaces. Of those who had breastfed in a public setting, two (n = 2; 10%) reported having difficulties finding a suitable public breastfeeding location, while one participant (n = 1; 5%) reported having no problems finding an appropriate place to breastfeed in public. Of those who had attempted to breastfeed in public, two participants (n = 2; 10%) reported having been stopped or made to feel uncomfortable while doing so.

Supporting information File 4 provides a detailed depiction of postnatal mothers' knowledge and attitudes towards breastfeeding using the Iowa Infant Feeding Attitude Scale (IIFAS)

scale [63]. Twenty (n = 20) postnatal mothers completed the IIFAS before being interviewed, yielding a mean score of 62 (SD 8.0). This mean score situated closer to the higher end of the IIFAS range (17–85), suggests a generally positive attitude towards breastfeeding among mothers.

A nuanced view of breastfeeding and formula feeding was observed with a significant number of the postnatal mothers, 85% (n = 17), indicating that breastfeeding enhances mother-infant bonding. Furthermore, 90% (n = 18) perceive breastfeeding as more economical than formula feeding, highlighting cost considerations as a potential factor for breastfeeding uptake. Regarding the practicality and social acceptance of breastfeeding, 80% (n = 16) indicated that formula feeding is preferable for mothers intending to return to work. Moreover, 45% (n = 9) found baby formula more convenient than breastfeeding, and 60% (n = 12) believe breastfeeding in public places is inappropriate, reflecting societal barriers to breastfeeding. Considering the nutritional needs and appetite of the infant, 50% (n = 10) indicated that formula-fed babies are at a higher risk of being overfed, whereas 55% (n = 11) disagree that breastfeeding leads to overfeeding. Nonetheless, there's a split in views about breastmilk's iron content, with 20% (n = 4) agreeing and 35% (n = 7) disagreeing that it lacks iron, pointing to varied beliefs about nutritional adequacy. Emotionally and relationally, 80% (n = 16) do not believe that formula-feeding mothers miss out on a significant aspect of motherhood. Similarly, 65% (n = 13) disagree that breastfeeding excludes fathers from father-child bonding.

## Findings

Four main themes were generated from the analysis of interviews with all participants: **1)** From Idealization to Informed Reality: Transforming Prenatal Breastfeeding Guidance **2)** Integrating Interactivity and Personalization for Enhanced Maternal Engagement **3)** Consistency and Quality of Information in Prenatal Breastfeeding Education **4)** Integrating Socio-Cultural Insights and Partner Involvement in Breastfeeding Education. The coding framework is provided in Table 4, and participant quotations substantiate each theme and subtheme (S6 Table), as demonstrated in the narratives below.

## 1) From idealization to informed reality: Transforming prenatal breastfeeding guidance

This theme focuses on the perspectives of postnatal mothers and healthcare professionals on the contrast between the aspirational portrayal of breastfeeding presented in the prenatal breastfeeding education class and mothers' actual experiences. Despite the education received, the majority of the postnatal mothers expressed feeling somewhat *"unprepared"* for breastfeeding. Although some of the postnatal mothers found the classes *"helpful,"* others shared that the classes *"fell short"* in supporting them through the spectrum of potential breastfeeding experiences, including breastfeeding difficulties that may arise;

*"I would, for the most part, they emphasized the positives, which was reassuring. But again, I would have liked to hear a bit more about the challenges, to balance things out. It's not all smooth sailing, so it's good to be prepared for any bumps in the road' (MR02).*

The postnatal mothers noted the partial acknowledgement of *"challenges"* during the classes and expressed a craving for practical guidance on overcoming breastfeeding challenges beyond just a cursory mention;

*"Yes, they did touch a bit on the challenges, but I felt it was a bit too idealistic at times. . . I would have liked some more discussion on how to manage those situations" (MG01).*

**Table 4. Overview of coding framework.**

| THEME TITLE | SUBTHEMES | ILLUSTRATIVE CODES |
|---|---|---|
| **Theme One**<br>From Idealization to Informed Reality–Transforming Prenatal Breastfeeding Guidance. | • Experiences of Unmet Educational Expectations in Shaping Breastfeeding Decisions and Self-Efficacy | • Idealistic portrayal of breastfeeding.<br>• Discrepancies between educational content and real-life experience.<br>• Balanced perspective and challenges. |
| | • Psychological Responses to Unexpected Challenges Following an Idealized Portrayal of Breastfeeding. | • Real-life challenges and expectations.<br>• Emotional support needs and psychological readiness.<br>• Unmet emotional support needs in breastfeeding education. |
| **Theme Two**<br>Integrating Interactivity and Personalization for Enhanced Maternal Engagement. | • Incorporating Interactive Elements into Virtual Classes. | • Q&A sessions.<br>• Feedback sessions.<br>• Breakout rooms for small group discussions.<br>• Enhanced interactive features.<br>• Interactive participation (Conversational engagement through shared experience). |
| | • Adapting Virtual Prenatal Breastfeeding Education to Reflect Individual Preferences Through Technology. | • Personalized virtual educational approaches<br>• Virtual engagement strategies.<br>• Innovative virtual engagement tools.<br>• Limitations of virtual engagement. |
| **Theme Three**<br>Consistency and Quality of Information in Prenatal Breastfeeding Education. | • Standardization and Coordination of Guidance for Enhanced Clarity | • Inconsistency in advice and information sources.<br>• Confusion arising from differing advice given by professionals.<br>• Mothers' reflections on inconsistent advice and how it shaped their confidence and breastfeeding decisions.<br>• Calls for a unified, standardized approach to breastfeeding education.<br>• Professional development and continuous training in lactation support.<br>• Recommendations for implementing consistent, evidence-based breastfeeding advice. |
| **Theme Four**<br>Integrating Socio-Cultural Insights and Partner Involvement in Breastfeeding Education. | • Socio-Cultural and Partner Dynamics in Breastfeeding Decisions. | • Desire for increased partner involvement in classes.<br>• Need for partner awareness on practical breastfeeding support.<br>• Importance of socio-cultural awareness in breastfeeding education.<br>• Engagement of partners in breastfeeding education. |
| | • Breastfeeding Confidence in Relation to Societal Norms. | • Challenges with public breastfeeding due to societal perceptions.<br>• Shifts in societal acceptance of breastfeeding in public settings.<br>• Desire for more discussion on navigating societal norms in classes.<br>• Societal stigma and pressure affecting breastfeeding confidence. |

The postnatal mothers also noted the discordance between the educational intent of the prenatal breastfeeding education classes and the pragmatic needs of mothers, expressing the desire for a *"more balanced approach."* Several postnatal mothers shared that while breastfeeding is often *"idealized"* in the prenatal breastfeeding classes as a *"perfect, serene bonding experience,"* the *"reality"* can be significantly more challenging, particularly in the early stages;

*"They painted breastfeeding as this perfect, serene bonding experience. But let's be honest, it can be tough, especially in the beginning. I wish they had prepared us more for the potential difficulties" (MR01).*

The postnatal mothers called for a curriculum that is *"more aligned"* with the complexities of breastfeeding, incorporating practical advice and strategies for overcoming common

difficulties. The postnatal mothers' perspectives emphasize the importance of setting realistic expectations and providing them with *"more depth and understanding."*

*"It seemed like they had a set curriculum, and they just went through the motions. There was a bit of talk about different challenges you might face, but it wasn't specific to whether you were a first-time mother or not. I felt a bit left out in that regard, like my concerns didn't matter as much"* (ML03).

The Healthcare professionals, on the other hand, offered a perspective that acknowledges this discrepancy but frames it within the context of their educational goals;

*"It's not like we don't discuss these challenges at all. We mention them, we may not discuss them in depth but we mention them in the class. But we don't just focus on it. It's never a priority because, the goal of the classes is to motivate mothers to choose to breastfeed"* (LC01).

**1.1- Experiences of unmet educational expectations in shaping breastfeeding decisions and self-efficacy.**   This subtheme addresses how the idealized portrayal of breastfeeding, often lacking in discussions around potential challenges, can lead to feelings of unpreparedness and disillusionment among mothers. Some postnatal mothers shared that while the educational content was informative, they felt it didn't encompass the *"holistic"* preparation they anticipated for a supportive breastfeeding experience. They described how *"inadequate knowledge"* about common breastfeeding challenges left them with *"unmet expectations,"* which they associated with a reduced sense of confidence in their ability to breastfeed and feelings of uncertainty about continuing;

*"Well, when I faced difficulties in the first few weeks, I was a bit discouraged. I remember the class focusing so much on how natural and beneficial it is, which it is, but I wasn't fully ready for the hurdles. It made me question my ability to continue breastfeeding and whether I was doing everything right"* (MG01).

The postnatal mothers described the initial postnatal period as *"quite stressful,"* associating the stress with the divergence between the class teachings and their actual experiences. For instance, one postnatal mother noted, *"I had set my expectations based on what was taught in the class,"* emphasizing that the unanticipated challenges *"shook her confidence"* and led her to *"question"* her ability to breastfeed;

*"To be honest, it made the first few weeks quite stressful. I had set my expectations based on what was taught in the class, so facing these unanticipated challenges made me question my ability to breastfeed. It definitely impacted my confidence and made me reconsider my decision to continue breastfeeding at times"* (MCM04).

Postnatal mothers pointed out that while the classes emphasized the *"naturalness and benefits"* of breastfeeding, they inadequately prepared them for the *"storm"* they faced. This lack of preparedness contributed to *"discouragement and self-doubt,"* particularly regarding their confidence in overcoming these challenges and their perceived preparedness to breastfeed;

*"Well, now that you mention it, I might've been a bit naive. The class was all sunshine, but nothing could've braced me for the storm that was coming. Soreness, the bab not latching*

*right off–it was a far cry from natural at the start. It knocked me sideways, so it did. Doubts crept in, wondering if I was cut out for this" (Mother L02).*

Some of the postnatal mother's narrative further underscores this sentiment, with one of the postnatal mothers expressing feelings of being *"left at sea"* regarding the practical challenges of breastfeeding, such as *"insufficient milk production and painful nipples."* She felt that the classes barely addressed these *"common issues,"* leaving her feeling *"let down"* and unprepared for the reality of breastfeeding. Some of the postnatal mothers with previous childbirth experiences shared their expectations for the class to explore specific obstacles and provide *"actionable guidance,"* particularly for women with *"previous unsuccessful breastfeeding experience"* who are going through childbirth again;

*"I was hoping, or maybe hoping, the class would touch base on some challenges and tips specifically for mothers like meself, who've done it before, faced challenges and might be facing different issues the second time" (MR01).*

**1.2 -Psychological responses to unexpected challenges following an idealized portrayal of breastfeeding.** This subtheme delves into the emotional and psychological responses of encountering unforeseen breastfeeding challenges among postnatal mothers due to the idealized information on breastfeeding presented in the classes. Most of the postnatal mothers noted a profound *"sense of overwhelm, frustration, upset, and anxiety"* upon encountering difficulties with breastfeeding that were not anticipated based on their prenatal breastfeeding educational preparation;

*" After the baby came, I was all at loss. The* breastfeeding *wasn't going as smoothly as I'd hoped, and I felt overwhelmed. I remembered the class and how it all seemed so positive there, but the reality was a lot different. It was frustrating, and I felt anxious because I wasn't prepared for the problems I was facing" (ML04).*

The postnatal mothers expressed a psychological burden of feeling *"isolated"* in their struggles, intensified by the perceived universality of positive breastfeeding experiences promoted during classes. The mothers illustrated how positive messaging, when not coupled with realistic portrayals of potential challenges, contributed to feelings of *"frustration"* and *"isolation,"* especially as new mothers;

*"Oh, for sure. When I started having trouble with breastfeeding, all those positive messages from the class made me feel like I was failing. I was frustrated and anxious because I wasn't prepared for this side of things. It felt like I was the only one struggling, which was isolating" (MR05).*

The postnatal mothers illustrated the emotional turmoil they experienced when the reality of breastfeeding challenges differed from the expectations set by the information they received during breastfeeding classes. They expressed a feeling of *"loss"* and *"anxiety,"* emotions rooted in the stark contrast between the anticipation of a *"beautiful, natural experience"* and the unforeseen difficulties encountered;

*"It was tough, really tough. I was all set for this beautiful, natural experience, but then, when we hit snags, I felt so lost and anxious. The sense of failure was overwhelming. If the class had*

*been more upfront about the potential difficulties and how common they are, maybe I'd have been kinder to myself when things didn't go as planned" (MC03).*

Most of the postnatal mothers talked about the lack of discussion regarding the mental and emotional obstacles of breastfeeding, such as *"postpartum depression."* The mothers expressed their emotional struggles. They believe that *"additional guidance and assistance"* could have supported their experience with emotional struggles;

*Well, it left me feeling quite anxious, you know? I wondered if I'd be able to handle it, especially as a new mom, or if I was the only one struggling. It's like I was set up to expect this beautiful, natural thing, but then hit with the reality that it's not always a walk in the park. It made me question whether I was ready or capable, which was frustrating" (MG05).*

The majority of the post-natal mothers voiced their expectations for *"more support"* and *"understanding"* in navigating the emotional landscape of breastfeeding. They sought discussions that addressed the *"frustrations and doubts"* inherent in the breastfeeding journey, particularly when not aligned with planned expectations. They yearned for a *"personal touch"* and reassurance to affirm their feelings were *"normal."* The mothers noted that psychological well-being is as crucial as physical preparation for breastfeeding;

*"Well, I suppose I was looking' for more support and understanding, like talking' about the frustrations and doubts a mother might have, especially if things don't go as planned. A personal touch, you know? A bit of reassurance that what I was feeling' was normal. The classes were very clinical, which is good in a way, but it didn't quite hit the mark for me" (ML01).*

Healthcare professionals acknowledged the gap in prenatal breastfeeding education. While reflecting on changes brought about by the pandemic in breastfeeding support, they emphasized the importance of emotional support in the breastfeeding journey and the need to balance practical guidance with emotional and psychological support;

*"You know, that's a very valid point, and it's something we've taken to heart. Historically, our focus may have been more on the physical aspects of breastfeeding, perhaps at the expense of the equally important emotional and psychological support. Breastfeeding is not just a physical act; it's a deeply emotional journey. With the pandemic forcing us to move our classes online, we've realized that we might have missed out on offering that crucial personal connection and emotional support" (GC02).*

As narrated by some of the lactation consultants, the transition to online breastfeeding support adds another layer to this complex issue, describing the online environment as sometimes feeling *"Impersonal."* In their opinion, this intensifies the sense of isolation and anxiety among mothers. They also acknowledged the challenges posed by the transition to online classes —*"Reduced engagement, limited hands-on learning, technological barriers, limited interaction"*—and reinforced their commitment to adapting their practices as a positive step towards better addressing the mothers' needs;

*"The shift online was both a blessing and a bit of a curse, to be honest. On one hand, it allowed us to keep providing essential information and support at a crucial time. But on the other, it's harder to replicate the warmth and personal connection of face-to-face interactions. The*

*nuances of comfort and understanding can sometimes get lost in translation over a screen. It's something we're keenly aware of and looking to improve upon" (RC03).*

The lactation consultants emphasized the importance of *"balancing"* prenatal breastfeeding educational content to include both the positive aspects of breastfeeding and realistic portrayals of challenges. Moreover, they noted the need for emotional support mechanisms, particularly in the digital landscape, to bridge the gap between *"expectation and reality."* Reflecting on the transition from face-to-face to online support, the lactation consultants underscored the necessity of innovative approaches to replicate the warmth and immediacy of face-to-face interactions, including using more interactive online tools;

*"The shift online was necessary, given the circumstances, but it's had its challenges, no doubt about it. While it's enabled us to continue providing critical information and support, the online format can sometimes feel impersonal. The physical cues and the immediate comfort you can offer in person are harder to replicate online. This can make it more difficult for mothers to express their concerns and for us to provide the nuanced support they need. We recognize this gap and are exploring ways to make our online sessions more interactive and supportive to better address these emotional aspects" (LC03).*

## 2) Integrating interactivity and personalization for enhanced maternal engagement

This theme focuses on the perspectives of postnatal mothers and healthcare professionals on the significance of tailoring prenatal breastfeeding education to meet expectant mothers' dynamic needs and preferences, thereby fostering a more engaging and supportive learning environment. Postnatal mothers desired a more *"interactive"* and *"personalized"* learning experience that delivers information and actively involves them in the learning process while acknowledging their unique circumstances. They expressed a preference for a platform that recognizes and values their individuality, encourages active participation, fosters a sense of community through shared experiences, and employs engagement strategies that are both *"sensitive"* and *"responsive"* to their needs;

*"But there's room for improvement. The class was a bit one-size-fits-all. I was hoping' for more opportunities to ask questions and get answers relevant to me own concerns. More interactive activities would've been nice, too, to really feel engaged and part of it all" (ML05).*

Most postnatal mothers talked about the essence of individualized breastfeeding support; *"Oh, it's essential! Every mum and baby pair are unique,"* noting the uniqueness of each mother-infant dyad. They discussed the importance of recognizing individual differences in an educational setting, underscoring the inadequacy of a *"one-size-fits-all approach,"* the value of shared experiences, and a shift from passive receipt of information to active participation. They expressed individuality and the desire for a conversational engagement that allows the mothers to not only be seen but also heard;

*"Oh, it's essential! Every mum and baby pair is unique. You see, it felt a lot like we were just being talked at, rather than being engaged in a conversation. As mothers, we've all got different experiences, different concerns, different situations" (ML01).*

The mothers also expressed the desire for active involvement; *"I mean, it would've been nice to feel more like a participant rather than just an observer, you know?"* reaffirming the importance of participation over mere observation and the need for educational practices that are not only informative but also engaging. Although some of them expressed *"optimism"* about the virtual format of the class, they conveyed a clear preference for interactive and personalized learning experiences that foster a sense of *"connection"* and direct applicability to their *"unique situations."*;

*"The class had great information, don't get me wrong, but it was mostly just watching and listening. There were no opportunities to engage directly, maybe through discussions or practical exercises that could've made it feel more personal and less like just watching a webinar"* (MR02).

**2.1-Incorporating interactive elements into virtual classes (e.g., Q&A sessions, feedback sessions, breakout rooms).** This sub-theme focuses on the perspectives of postnatal mothers and healthcare professionals on integrating interactive features into virtual learning environments. Most post-natal mothers expressed a desire for formalized feedback mechanisms to share their thoughts on the classes anonymously. They described the lack of such mechanisms as a *"missed opportunity"* for educators to improve the class experience; *"To be honest with ye, there wasn't any formal way for us to give feedback after the class."* The mothers equally highlighted the importance of anonymous feedback mechanisms to improve course content delivery and enhance the learning experience;

*"My biggest suggestion would be to introduce a simple, anonymous way for us mums to give feedback on each class whether it's through an online form or a suggestion box kind of thing, just something that lets us voice our thoughts without worry. And then, of course, for that feedback to be taken on board and used to make the classes even better"* (ML04).

Healthcare professionals also recognize the importance of creating a supportive and responsive educational environment. They expressed *"openness to innovations"* such as anonymous feedback mechanisms, which could inform adjustments to better meet mothers' needs;

*"We always aim to create an environment where mothers feel supported and heard. The idea of an anonymous feedback mechanism hadn't crossed our minds, but it's certainly something we should consider. It would give us a chance to learn what's working well and where we might need to make adjustments. It's all about making sure the mums and their families have the best possible experience and support"* (GC02).

Considering the uniqueness of the mother's needs, most postnatal mothers strongly preferred *"personalized guidance"* through one-on-one sessions or smaller group discussions. They suggested that smaller, more personal discussion groups or one-on-one sessions would help address *"unique questions and concerns."* They felt such formats could better support *"specific concerns"* compared to larger class settings;

*"Ok, let's look at it this way, some mothers might prefer face-to-face interactions or need more in-depth support that's better provided in person. So, while over-the-phone consultations are beneficial, a combination of options, including in-person and small-group sessions, could*

*better address the diverse needs of all mothers. I mean we still had to go in to the hospital to see our doctors so...” (MR01).*

The healthcare professionals also acknowledged the importance of creating a supportive and personalized educational environment for expectant mothers. They discussed the potential benefits and the value of personalized guidance. However, they noted the challenges imposed by COVID-19 restrictions that have led to adjustments, such as moving sessions online or enlarging class sizes. They expressed how this move from in-person classes to virtual platforms has limited the ability to offer personalized interaction and support and shared the desire to improve in this area. They also underscored the importance of more intimate support structures and the inclusion of real-life stories and practical advice to make the classes more *relevant* and reassuring for mothers;

*“Well, that's the thing. Personalized guidance is something we value a lot, but with the Covid restrictions and all, we've had to hold back on how much of that we can offer. It's been tough, not being able to provide the one-on-one support we know is so beneficial. We've had to move a lot of our sessions online or make them larger, which isn't ideal. It's a challenge, for sure, but we're hoping to get back to more of that personal touch when we can” (CC03).*

The postnatal mothers critiqued the classes for their *“passive nature,”* preferring more active forms of engagement; *“I was just sitting there watching a screen without much chance to get involved.”* Although the mothers said the classes were informative, they also expressed frustration with the lack of interactive opportunities, indicating a clear preference for a more engaged and participatory learning environment;

*“The class had great information, don't get me wrong, but it was mostly just watching and listening. I would've loved more opportunities to engage directly, maybe through discussions or practical exercises that could've made it feel more personal and less like just watching a webinar” (MG02).*

Some postnatal mothers acknowledged efforts to incorporate interactivity through polls and quizzes. However, they felt the interactive elements did not provide the engaging learning experiences necessary to support their breastfeeding journeys. They noted the need for advanced interactive tools, such as *“breakout rooms,”* for discussions and one-on-one interactions with lactation consultants/midwives. They described these tools as essential for making their learning experience more *“immersive”* and *“interactive.”*

*“In a topic as hands-on as breastfeeding, you really need to be able to interact more directly. The polls and quizzes were grand for a start, but they didn't allow for back-and-forth conversation. There were times I had questions or wanted to hear more about others' experiences, but the format just didn't support that kind of interaction. It's all very well knowing the theory, but breastfeeding is practical, isn't it?” (MG01).*

Most postnatal mothers expressed a preference for a prenatal breastfeeding educational model that transcends traditional didactic approaches, and they talked about the value of *“shared experiences.”* They noted the importance of hearing other mothers' experiences, suggesting that real-life stories and practical tips could provide reassurance, a sense of relatability, and a deeper understanding that goes beyond *“textbook”* information;

*"I mean, hearing other mums share their experiences would have provided real-life stories and practical tips that go beyond the textbook information presented in the class. Sometimes, knowing that someone else has faced similar challenges and succeeded can be incredibly reassuring" (MG03).*

Further underscoring the importance of participatory learning and shared experience, the mothers suggested that leveraging the collective knowledge and experiences of other mothers in the classes can complement the expertise provided by the professionals, offering a richer, more diverse learning experience that benefits first-time mothers as well as those with prior experience. They noted the importance of giving voice to all participants and recognizing the wealth of knowledge they bring;

*"You can imagine sitting down in front of your screen there for nearly two hours listening to someone speak. I think there are other ways to make the classes more interesting. Like watching the videos, I told you about was good, I also think that it would have been more interesting to hear from other moms who had breastfed or had a different point of view. It felt more like a lesson than a workshop where people".*

*could talk to each other. I think the class would have been more interesting and easier to understand if there had been a mix of views and experiences" (MR05).*

The postnatal mothers also raised concerns about the structured nature of the setup, noting that it discouraged direct questioning and encouraged passive participation through chat boxes. They pointed out the importance of real-time dialogue over indirect communication channels like chat boxes, expressing the need for more fluid, adaptive learning environments that encourage spontaneous interactions and discussions;

*"I would say it was more of an informal arrangement, if you know what I mean. We weren't told not to ask questions directly, but the way the class was set up made it clear that we should ask questions in the chat box. It would have been more interesting if people could talk to each other in real timely. Then again, it was quite a large class with so much to cover all at once. With the way the class was structured it would seem like an interruption to unmute and try to ask a question. I suppose the more convenient thing to do was just to type in the question un the chat box"(MC03).*

The healthcare professionals acknowledged the challenges posed by the interactive approach, which aims to provide *"real-time support."* They explained that the need for having two consultants—one to teach the class and the other to answer questions in the chat box—supports efficiency. Still, it might feel overwhelming for mothers, especially first-time mothers trying to absorb significant information. Furthermore, they also recognized that although some mothers might find the chat box system convenient for receiving immediate responses, it may not be suitable for everyone. They observed differences in mothers' comfort levels with technology and noted that busy chat activity could lead to some questions being overlooked;

*"Ah, yes, that's a fair point. We do have two consultants for efficiency, but I see how it can be a bit overwhelming for the mothers listening and having to type in their questions in the chatbox. The idea is to give real-time support. While one of us is talking, the other answers questions so there's no wait. But I understand it can be distracting, especially for first-time mums who are trying to absorb a lot of information" (LC01).*

**2.2-Adapting virtual prenatal breastfeeding education to reflect individual preferences through technology.** This subtheme reflects the perspectives of mothers and healthcare professionals on adapting virtual prenatal breastfeeding education to incorporate innovative strategies that empower mothers with the knowledge and confidence to make informed decisions about breastfeeding. The classes were mostly held on virtual platforms, and the mothers described the innovative potential of *"leveraging technology"* to deliver information and create a more personalized, engaging, and supportive learning environment. The mothers suggested a *"forward-thinking approach"* to prenatal breastfeeding education that could better support mothers' breastfeeding decisions by addressing their collective concerns. They also suggested the introduction of *"digital platforms"* and innovative solutions utilizing *"virtual reality scenarios"* to simulate hands-on breastfeeding techniques. The mothers proposed *"rethinking"* the delivery of prenatal breastfeeding education to include tracking learning progress, setting reminders for practicing techniques, and logging their breastfeeding journeys;

> *"I think it would be beneficial to have some one-on-one sessions available. Even if it's just a short video call with the lactation consultant to discuss personal concerns or questions. Group sessions are informative, but individual advice could make a big difference. Also, perhaps a virtual buddy system with other expectant mothers to share experiences and support each other"* (MG03).

Further reinforcing the need for an innovative prenatal breastfeeding educational approach, the mothers emphasized the value of *"personalized learning paths"* tailored to individual concerns and learning styles, as determined by a *"pre-class survey."* Postnatal mothers equally described the concept of a pre-class survey: *"The pre-class survey I imagined would be a detailed questionnaire sent to participants before the class starts. It would ask about their current breastfeeding knowledge, specific concerns, learning preferences, and emotional state regarding breastfeeding."* They suggested that this approach would shift away from one-size-fits-all education towards more *"adaptive models"* that address the specific needs of each mother. For example, offering more resources and simulations focused on latching for those feeling anxious about it;

> *"I think personalized learning paths could make a big difference. Based on a pre-class survey about our concerns and learning styles, the class could offer tailored content. For instance, if a mother is particularly anxious about latching, she would receive more resources and simulations focused on that. Also, integrating a virtual support group feature where you can instantly connect with other mothers or a consultant for live advice could provide immediate reassurance and a sense of community"* (MG04).

On the other hand, healthcare professionals acknowledged the need to enhance educational tools, mainly through online demonstrations focusing on crucial techniques like latching. They noted the multifaceted nature of learning, suggesting a blend of digital innovation with the irreplaceable value of direct human interaction;

> *"I would say that technology is ever-evolving, and yes, we do hope for enhancements that could provide a more "hands-on" virtual experience in the future. Even at that, the human aspect, the emotional and physical support that comes from being in the same room, might always be a missing piece in online education and support"* (LC01).

The healthcare professionals also suggested adopting a *"hybrid model"* that combines online education with face-to-face sessions: *"We're looking into combining both online and face-to-*

*face classes that combine online education with face-to-face sessions."* They noted that this approach would allow for personal interaction and hands-on practice, which can be reassuring for expectant mothers. Healthcare professionals described the hybrid model as a valuable solution to combine the strengths of both digital and traditional educational methods;

> *"A hybrid model could offer the best of both worlds. While online classes provide convenience and accessibility, adding face-to-face sessions could foster a deeper connection and allow for real-time feedback. We're exploring options for small, in-person workshops that comply with safety guidelines, where mothers can practice techniques and ask questions in a supportive setting" (LC03).*

The collective viewpoint of the healthcare professionals indicated that this approach assumes a level of digital literacy and access that may not be universal, posing potential barriers to inclusivity. They noted the challenge of seamlessly integrating these modalities to ensure that each complements the other without increasing disparities in access or engagement;

> *"We're aware that not everyone has the same access to technology or feels comfortable in online settings. I personally think that the best way to address this is to offer resources in various formats and ensure our in-person sessions are small and welcoming. This way, we hope to reach as many mothers as possible without creating disparities in access or engagement" (LC01).*

### 3) Consistency and quality of information in prenatal breastfeeding education

This theme focuses on the perspectives of postnatal mothers and healthcare professionals regarding the standardization and quality of prenatal breastfeeding education in Ireland. It reflects the need for consistency and accuracy in the information delivered to expectant mothers and notes the potential implications of prenatal classes on breastfeeding decisions. The postnatal mothers noted the challenge of navigating the *"differing opinions"* presented by three lactation consultants on key breastfeeding practices, such as the frequency of feeds and optimal latching positions. For example, one mother described a pivotal moment of adaptation when she was left with no option but to *"focus on what felt right for her and her baby"* and expressed that, while inconsistent advice can be confusing, it may encourage mothers to trust their instincts and build confidence in their breastfeeding decisions;

> *"I found the advice helpful, but sometimes it got a bit confusing. The lactation consultants seemed to have slightly different opinions on some matters, like how often to feed or the best positions for breastfeeding" (GM01).*

One mother illustrated the difficulty caused by receiving inconsistent breastfeeding advice by comparing it to *"cooking stew,"* where *"everyone's got their own recipe,"* their way of doing it. The mother described how the varied guidance led to *"head spinning,"* prompting her to evaluate the advice and personalize her approach critically. Notably, she reflected on how this situation led her to introspection, saying she *"had a sit-down with myself"* to sift through the conflicting information and embrace what resonated with her personal beliefs and circumstances;

> *"There was a wee bit of that. It's a bit like when you're cooking a stew, isn't it? Everyone's got their own recipe, their own way of doing it. So, when one nurse tells you one thing and another*

*says something else, it can leave your head spinning. It gave me pause, so it did. But, you know, I had a sit-down with myself. Thought about all the chatter and decided to sift through it, take what made sense to me" (MR04).*

Postnatal mothers described their experience as *"tricky and confusing,"* with differing advice leading to *"second-guessing."* They expressed the importance of direct, individualized communication to address and resolve the confusion arising from inconsistent advice and the role of healthcare professionals in providing clear, consistent guidance tailored to individual needs;

*"Well, that's a tricky one. At times, it was a bit confusing, to be honest. One person would say one thing, and then another would say something a bit different. It made me second-guess myself at times. But I talked it over with the lactation consultant after one of the classes, and she was able to clarify things for me" (MR01).*

**3.1- Standardization and coordination of guidance for enhanced clarity.** This subtheme focuses on the perspectives of postnatal mothers and healthcare professionals on the standardization and quality of prenatal breastfeeding education in Ireland. It reflects the need for consistency and accuracy in the information delivered to expectant mothers and considers the potential role of prenatal classes in shaping breastfeeding decisions. This theme also examines the variability in competence among healthcare professionals, which can affect the quality and helpfulness of breastfeeding education for mothers. Most of the postnatal mothers expressed frustration with the lack of consistency in advice, noting that the confusion they felt was a matter of logistical inconvenience and a significant emotional burden, leading to *"self-doubt"* and a diminished sense of *"self-efficacy;"*

*"But then, you'd get a bit of advice from one professional, and something quite different from another. It was confusing, to be honest. It threw me for a loop, it did. Here I was, trying to do the best for my wee one, and the mixed messages were stressing me out"(ML05).*

However, the healthcare professionals addressed the inconsistency issue from a professional perspective: *"Inconsistencies can arise due to varying personal experiences and training backgrounds among professionals."* The healthcare professionals acknowledged the diversity in training and experiences as significant contributors to the lack of uniformity in breastfeeding guidance, stressing the need for more precise, more unified guidance and recognizing the confusion and uncertainty that inconsistency can create for mothers;

*Yes, I'm aware of this issue. The diversity in training and experiences among healthcare professionals can lead to differences in the breastfeeding advice we give. While we all aim to provide the best guidance, these variations can sometimes confuse mothers. It's something that should really be worked on (LC01).*

The postnatal mothers proffered the solution of a *"unified approach"* to breastfeeding education, with one stating, *"They need to be on the same page. The professionals do. If they could just give consistent advice, it would make a world of difference."* They described this uniformity as a means to address confusion and stress, emphasising that consistency in guidance could positively shape mothers' breastfeeding experiences. Furthermore, they described how the consequences of inconsistent advice extend beyond individual experiences, fostering a shared sense of confusion among mothers. One mother cited variation in recommendations on how

often to feed the baby as a concrete example of the confusing advice that intensified mothers' feelings of uncertainty;

> *"One consultant would be saying one thing about how often to feed the baby, and another would have a slightly different take. It was a bit confusing at times. You're there, trying to learn the best you can for your wee one, and when you get mixed messages, it's hard to know what's best" (MC05).*

The solution to this issue, as proposed by the healthcare professionals, involves the *"standardization"* of training across all healthcare professionals involved in prenatal care. Healthcare professionals unanimously indicated that standardization is essential for supporting all professionals, regardless of their role, in being equipped with the knowledge and skills to provide empathetic and supportive breastfeeding education;

> *"There needs to be a systems approach. The same checklist, same learning outcomes across board. The mother is entitled to get the same standard in all hospitals" (GC03).*

The healthcare professionals also recognized the challenges of inconsistent advice for mothers; *"Inconsistent advice can certainly throw a spanner in the works. It can confuse the mothers, making them second-guess their choice to breastfeed."* They reflected on a deep understanding of the anxiety and uncertainty that inconsistent guidance can create and emphasized that the overarching goal is to *"empower mothers,"* not increase their anxiety, while diligently striving to provide consistent and clear advice. A consensus opinion of the mothers reflects a more *"coordinated and consistent approach"* to breastfeeding education, emphasizing the necessity of clear, unified guidance to support mothers in their breastfeeding journey. Their respective narratives echoed the urgent need for a standardized approach to prenatal breastfeeding guidance, advocating for clarity and accuracy from healthcare professionals;

> *"I think a more coordinated approach to the information provided would be beneficial. Maybe having a section of the class where they address common misconceptions or differences in advice could help. Also, it would be great if they could provide some resources for further reading, where the information aligns with what was taught in the class" (ML05).*

On the other hand, the healthcare professionals pointed out a *"systemic"* issue affecting the quality of prenatal breastfeeding education, drawing attention to the importance of enhanced communication, coordination, and standardization among healthcare professionals to support the delivery of consistent and evidence-based breastfeeding education;

> *"Another challenge is ensuring communication and coordination between lactation consultants and midwives. All staff must be on the same page and working together to provide consistent, evidence-based support to mothers" (GC01).*

Additionally, the lactation consultant highlighted the importance of regular workshops and seminars and creating a supportive network where professionals can discuss cases and share advice to maintain a *"consistent approach"* to breastfeeding education. They also emphasised the importance of continuous professional development initiatives to keep healthcare professionals abreast of the latest breastfeeding practices and research, supporting a consistent approach to prenatal breastfeeding education;

*"Regular workshops and seminars could also help keep everyone updated on the latest breast-feeding practices and research. Additionally, creating a supportive network where profession-als can discuss cases and share advice would help maintain a consistent approach. Again, providing ongoing training and education for lactation consultants and midwives on best practices for supporting breastfeeding mothers is essential. This can help ensure that all staff are current on the latest research and recommendations for breastfeeding support"* (CC03).

## 4) Integrating socio-cultural insights and partner involvement in breastfeeding education

This theme focuses on the perspectives of postnatal mothers and healthcare professionals on the importance of fostering a supportive environment for breastfeeding through enhanced socio-cultural understanding and active partner involvement in prenatal breastfeeding educa-tion classes. The postnatal mothers expressed a desire for greater partner involvement, observ-ing that while partners were welcome to attend prenatal breastfeeding classes, it was not *"strongly emphasized"* in the classes they attended. Despite the acknowledgement of partners being welcome in classes, the mothers pointed out the lack of emphasis on their attendance as an *"overlooked potential"* for enhancing support for breastfeeding;

*"The classes were focused on the mother-baby relationship and how to breastfeed. But I think it would have been really beneficial for my partner to attend the classes with me"* (MR05).

**4.1- Socio-cultural and partner dynamics in breastfeeding decisions.** This subtheme highlights the views of postnatal mothers and healthcare professionals on the significant impact of partner involvement in breastfeeding classes. Most postnatal mothers believed that the direct participation of their partners in these classes would have enhanced their partner's understanding and ability to support the breastfeeding process. They illustrated the perceived value of first-hand knowledge and engagement of partners;

*"Yeah, well, my partner was really supportive, but I think it would have helped him to under-stand the process better and know how he could help me. Plus, he would have been able to ask questions and get more involved in the process"* (MR01).

The healthcare professionals' view on partner involvement complemented the postnatal mother's perspective. They emphasized the critical need to equip partners with practical skills and knowledge, fostering a more empowered and collaborative approach to breastfeeding sup-port; *"Well, partners can be a huge help in many ways. . . It's about making decisions together as a team"*. Moreover, the postnatal mothers express a shared understanding of the challenges associated with breastfeeding, highlighting that while partners are supportive, their lack of knowledge on how to assist practically can be a significant barrier. They asserted the critical need for educating partners not just on the importance of breastfeeding but also on practical ways they can contribute to easing the mother's burden;

*"My partner was supportive, but he didn't really know how to help me with breastfeeding. My mother and sister both breastfed their children and were helpful with advice, but I felt like they didn't really understand how challenging it could be for me"* (MG03).

Furthermore, the healthcare practitioner's perspectives reinforced the notion that breast-feeding success is closely linked to a robust support system including partners. They

acknowledged breastfeeding is a *"team effort"* and the importance of involving partners *"as much as possible"*. This aligns with the mothers' desires for more *"inclusive"* breastfeeding educational practices. Yet, the lactation consultants also note a *"shortfall"* in partner attendance, indicating a potential area for improvement in how these classes are structured to encourage partner participation better;

> *"We've found that mothers who are highly motivated and committed to breastfeeding tend to have the most success, regardless of their individual circumstances. We also see higher rates of success among mothers who have a strong support system, whether that's a partner, family member, or lactation consultant"* (LC01).

Both postnatal mothers and healthcare professionals agreed on the significance of the *"socio-cultural"* insights in breastfeeding education. The postnatal mothers specifically expressed the desire for broader *"community support"* and recognition of breastfeeding as a *"shared responsibility"*. They emphasized that societal perceptions and norms play a substantial role in shaping individual breastfeeding experiences. Collaboratively, the healthcare professionals reinforced this idea by demonstrating professional awareness of the socio-cultural factors in shaping breafeeding breastfeeding practices, aiming to *"support not just the mammy but the whole family"*. This aligns with an assertion by one mother that *"breastfeeding isn't a solitary task, it's a family affair"*, speaking to the need for a cultural shift towards more inclusive prenatal breastfeeding educational support for mothers;

> *"If there were more understanding and support in the community, then breastfeeding mothers might feel less isolated and under pressure. Also, it might help if more people understand that it's not only the mum's job to feed and nurture the baby, but it's also important for partners and family members to take active roles in supporting and participating in this process"* (GM03).

**4.2-Breastfeeding confidence in relation to societal norms.** This subtheme explores the challenges postnatal mothers face with breastfeeding in public and healthcare professionals' perspectives on discussing this topic during classes. Despite a perceived shift toward greater societal acceptance, it indicates the absence of explicit discussion in education classes. The postnatal mothers highlighted the lack of specific discussion on *"breastfeeding in public"* during the classes. Although the mothers noted a *"shift towards more acceptance"* of public breastfeeding and indicated a changing societal landscape, they emphasized the need for explicit support and guidance to navigate *"societal attitudes"* towards it in public. They reiterated the need for openness and understanding, pointing to the essential role of societal support in fostering a *"breastfeeding-friendly"* environment;

> *"No, the classes didn't specifically address breastfeeding in public, but I do think it would be helpful for future classes to include information on this topic. I think there's definitely been a shift towards more acceptance of breastfeeding in public in recent years in this country. I also think it's very important for people to be more open to and understanding of breastfeeding in public"* (MR02).

Some of the postnatal mothers who had attempted breastfeeding in public narrated their experience of breastfeeding in public, revealing a series of reactions from onlookers, ranging from indifference to overt staring. They highlight the varied societal responses to breastfeeding

and mothers' challenges in balancing natural motherhood practices with societal perceptions, underscoring the conflict between individual confidence and societal norms;

> *"Yes, I have breastfed my baby in public before, and people's reactions were varied. Some people appeared unconcerned, while others just stared. It can be difficult to breastfeed in public, I believe that breastfeeding is a natural and necessary aspect of motherhood, and that women should not be made to feel humiliated or for breastfeeding in public" (CM04).*

A typical example of the postnatal mother's experience with breastfeeding in public is captured in one of the mother's expressions of her pre-existing determination to breastfeed; "*I was set on breastfeeding anyway*". However, she also acknowledged the potential impacts of societal pressure on her decision to breastfeed, which made her feel *"already a bit unsure"*. She highlighted the importance of more discussions on *"real-life experiences of breastfeeding in public"* and the need for practical advice beyond the reassurance that *"it's legal,"*. She suggests that legal knowledge alone is insufficient to build confidence against societal scrutiny;

> *"I was set on breastfeeding anyway. But I think it might affect some women, especially those already feeling a bit unsure. And even for me, it would have been comforting to have more discussion about the real-life experiences of breastfeeding in public, rather than just the 'you can do it, it's legal' part, you know?" (CM04).*

The healthcare professionals acknowledged the importance of discussing public breastfeeding in the classes, particularly in the *"Irish context"*, where a discrepancy exists between high initiation and low continuation rates. They identified *"societal challenges"* as a significant barrier to sustained breastfeeding;

> *"I definitely think it's important to discuss breastfeeding in public in these classes, especially in the Irish context. Ireland has a relatively high breastfeeding initiation rate, but a low rate of breastfeeding continuation, which suggests that many mothers may be facing challenges with breastfeeding in public. By discussing breastfeeding in public and addressing any concerns or fears that mothers may have, we can help to empower them to breastfeed confidently and comfortably in any setting" (RC02).*

The postnatal mothers drew the comparison between the support for breastfeeding within their network and the *"awkwardness and stigma"* associated with breastfeeding in public. They described the internal conflict that mothers may face, torn between the support of their immediate circle and the broader societal pressures, potentially undermining confidence in breastfeeding publicly;

> *"In me own circle, there's a fair bit of breastfeeding support. Many of the mums I know, they've breastfed their wee ones and they're all for it. But, there's also a good few who find it a bit awkward, you know, doing' it in public or around other people. There's still a bit of a stigma attached to it, like. Not everyone's comfortable with it" (RM04).*

Considering the acceptance of breastfeeding in public within the Irish context, the healthcare professionals acknowledged the challenges but noted a shift towards greater acceptance, suggesting *that "Irish society as a whole is changing and becoming a more accepting place for breastfeeding mothers."* However, they expressed an optimistic view of societal evolution, albeit recognizing the current complexities of breastfeeding in public as informed by *"both national norms and personal views."*

*"Breastfeeding in public is a complicated issue, especially here in Ireland. that is affected by both national norms and personal views, but I think that the Irish society as a whole is changing and becoming a more accepting place for breastfeeding mothers" (LC03).*

From the healthcare practitioner's perspective, the lactation consultants acknowledged breastfeeding mothers' multifaceted challenges, including *"lack of support from family and friends, returning to work, and difficulty breastfeeding in public"*. Their emphasis on the pressure on mothers to formula feed *"due to cultural or societal expectations"* aligns with the mothers' concerns about societal attitudes. The healthcare professionals highlighted their role in *"educating and empowering mothers to make informed decisions"*, advocating for their rights to breastfeed in any setting and overcoming the barriers posed by societal norms;

*"One of the most common deterrents to breastfeeding in Ireland is the lack of support for breastfeeding in public places. Although Limerick city is breastfeeding friendly, but we still have a lot to do in this area in form of advocacy and creating awareness, I mean the government can do better because many mothers feel uncomfortable or judged when they try to breastfeed in public, which can lead them to avoid breastfeeding altogether" (LC03)*

## Discussion

This study explored midwives'/lactation consultants' and mothers' perspectives on prenatal breastfeeding education classes in Ireland and how their implications on the mothers' breastfeeding decisions. Postnatal mothers in this study expressed a profound sense of unpreparedness for breastfeeding despite attending prenatal breastfeeding education classes. They indicated that these classes, while providing essential theoretical knowledge, often fall short of equipping mothers with the practical skills necessary to manage breastfeeding challenges. This perception suggests that BF classes may disproportionately emphasize the benefits of breastfeeding while neglecting to address potential challenges adequately. Findings from a study conducted in Sweden demonstrate that many women expect breastfeeding to be instinctive, easy, and pleasurable [68]. However, the experience often diverges significantly from these expectations, with new mothers frequently unprepared for the considerable challenges of initiating and sustaining breastfeeding [69, 70]. Healthcare professionals in this study acknowledged the discrepancy between the objectives of prenatal breastfeeding classes and the actual experiences reported by mothers. They justify their current BF classes by underscoring their primary aim: promoting breastfeeding initiation. This approach prioritizes the benefits of breastfeeding to encourage breastfeeding initiation, albeit potentially at the expense of adequately addressing potential challenges. However, postnatal mother feedback suggests a pressing need to reassess this strategy to better align with their needs and expectations. This finding corresponds with the conclusions of a scoping review [71], which sought to identify mothers' obstacles when breastfeeding and the support required to facilitate their decisions. Commonly cited phrases such as "breast is best" and "breastmilk is best" were frequently mentioned by study participants in the selected studies. Paradoxically, the emphasis on the importance and perceived ease of breastfeeding contributed to a disconnect between women's expectations and their actual breastfeeding experiences.

This study revealed that postnatal mothers frequently experience a range of negative emotions, including feeling overwhelmed, frustrated, anxious, and isolated when facing breastfeeding difficulties. The lack of sufficient focus on the mental and emotional aspects of breastfeeding, such as postpartum depression and anxiety, intensifies the psychological burden

on new mothers. Postnatal mothers emphasized that emotional support is as critical as educational support in positively influencing the decision of mothers to breastfeed and the overall breastfeeding experience for mothers. Positive breastfeeding experiences are associated with strong emotional support, irrespective of the quality of educational support [72]. Conversely, negative experiences are more prevalent when both emotional and educational support are insufficient. Healthcare professionals in this study recognized the imperative to address both the physical and emotional dimensions of breastfeeding holistically. Nevertheless, the transition to online classes necessitated by the pandemic exacerbated the challenges of providing adequate emotional support.

While prenatal breastfeeding classes provided essential knowledge, many mothers felt these sessions failed to meet their specific needs and lacked active participation; therefore, postnatal mothers preferred interactive and personalized learning experiences. This study highlights the importance of integrating interactive elements into virtual prenatal breastfeeding classes, as postnatal mothers and healthcare professionals have recommended. These elements include Q&A sessions, feedback mechanisms, and breakout rooms for small group discussions, facilitating direct engagement, enabling mothers to ask questions, share experiences, and receive real-time support. This strategy is corroborated by systematic reviews [24, 43] demonstrating the effectiveness of interactive antenatal education combined with postnatal web-based support in enhancing breastfeeding outcomes. Furthermore, previous studies [73, 74] highlight that incorporating real-life narratives from mothers who have navigated similar challenges can provide reassurance, normalize diverse breastfeeding experiences, and mitigate feelings of isolation and inadequacy. This stress on the importance of interactive and personalized learning experiences should make the audience feel the need for change in the current approach.

The transition to virtual platforms for prenatal breastfeeding education, driven by the COVID-19 pandemic, presents challenges and opportunities. Postnatal mothers in this study emphasized the potential of technology to enhance educational experiences, recommending digital platforms and virtual reality scenarios to address common concerns and influence breastfeeding decisions. This is consistent with the findings of [25] in response to the cessation of in-person prenatal education services during the COVID-19 pandemic. Additionally, mothers proposed personalized learning paths based on pre-class surveys to adapt to individual preferences, forming an adaptive educational model. Findings in a systematic review [30] demonstrate that a flexible prenatal breastfeeding educational framework effectively addresses mothers' specific needs through personalized experiences. Complementarily, healthcare professionals in this study advocate for a hybrid model that combines online education with face-to-face sessions and aligns with postnatal mothers' preferences. This approach offers convenience and accessibility while preserving essential personal interaction and hands-on practice for effective learning. Research supports the benefits of hybrid learning in healthcare education [75, 76]. However, this model requires a digital literacy and access baseline, which may not be universally available, potentially limiting inclusivity [77].

In this study, both postnatal mothers and healthcare professionals highlighted the detrimental effects of inconsistent breastfeeding advice. Mothers reported that receiving contradictory information from various professionals was not only logistically inconvenient but also emotionally taxing, leading to self-doubt and diminished self-efficacy. Other studies [78, 79] support this, indicating that standardised training improves the quality and consistency of healthcare delivery. However, healthcare professionals in this study acknowledged that these inconsistencies often stem from diverse personal experiences and training backgrounds.

This study identified the necessity of incorporating socio-cultural insights and partner involvement into prenatal breastfeeding education. Both postnatal mothers and Healthcare professionals recognize the profound impact of partner participation and socio-cultural

understanding on breastfeeding outcomes. Previous studies [80, 81] highlighting the critical role of partner involvement in achieving successful breastfeeding outcomes corroborate this finding. Moreover, research has demonstrated a significant correlation between antenatal education and fathers' antenatal attachment to the foetus [82]. Postnatal mothers in this study emphasized the need for broader community support and the recognition of breastfeeding as a collective responsibility, noting that societal perceptions and norms substantially influence individual breastfeeding experiences. In addition to physical and psychological breastfeeding challenges, negative societal attitudes were identified as a significant barrier to positive breastfeeding decisions [83, 84]. Although postnatal mothers observed a trend towards greater societal acceptance of breastfeeding in Ireland, they still required explicit support to navigate prevailing attitudes. Mothers who breastfed in public encountered a range of responses, from indifference to overt staring, illustrating the tension between breastfeeding confidence and societal norms. Despite their determination, some mothers felt that societal scrutiny eroded their confidence. Healthcare professionals reiterated this perspective, emphasizing the importance of understanding the sociocultural factors influencing breastfeeding practices in Ireland.

## Recommendation

To mitigate the unrealistic expectations often fostered by the idyllic portrayal of breastfeeding in prenatal education, realistic scenarios that discuss common challenges and offer practical solutions are imperative. Revising educational content to balance the positive aspects of breastfeeding with potential challenges and additional training for healthcare professionals will ensure comprehensive guidance. The study identified a critical need for enhanced guidance and support to address the negative emotions experienced by many postnatal mothers due to breastfeeding challenges. This underscores the importance of integrating emotional resilience and mental health support into prenatal education. Such integration should include information on recognizing and managing postpartum depression and providing access to psychological support resources.

Furthermore, postnatal mothers prefer adaptive learning environments that encourage spontaneous interactions and discussions, highlighting the importance of real-time dialogue in fostering an engaging and supportive learning experience. Therefore, prenatal breastfeeding education programs should incorporate flexible and interactive elements that facilitate live discussions, enabling mothers to share their experiences and receive immediate feedback and support. The findings also indicate that interactive online formats, such as video conferencing and virtual reality scenarios, effectively provide tailored support, timely information, and personalized guidance. Additionally, postnatal mothers recommended using digital tools to track learning progress, set practice reminders, and log breastfeeding journeys. Incorporating these interactive online formats and digital tools into prenatal breastfeeding education will enhance support, engagement, and personalized learning for mothers. Moreover, standardized training for healthcare professionals is essential to ensure they possess current knowledge and skills in breastfeeding education. This training will ensure consistent and accurate breastfeeding guidance. Implementing a comprehensive training program alongside ongoing professional development is recommended to support and educate mothers effectively.

Additionally, broader community support and acknowledging breastfeeding as a collective responsibility are crucial. Societal perceptions and norms significantly impact individual breastfeeding experiences. Mothers consistently desired increased partner involvement in prenatal classes, noting the inadequacy of emphasizing the active engagement of partners despite their invitations. Healthcare professionals must proactively integrate prospective fathers into antenatal breastfeeding education programs to address this gap. The study demonstrates that

mothers breastfeeding in public encountered diverse reactions, from indifference to overt staring. This range of responses underscores the conflict between maternal breastfeeding confidence and the societal norms they navigate. Therefore, Interventions promoting behavioural change regarding breastfeeding should prioritize eradicating beliefs and practices that contribute to suboptimal breastfeeding outcomes while reinforcing positive behaviours.

## Strengths and limitations

A notable strength of this study is its originality, marking the inaugural exploration of perspectives on prenatal breastfeeding education within the Irish context. This research significantly contributes to the existing literature by offering insights into the experiences of midwives, lactation consultants, and postnatal mothers, facilitating a multifaceted understanding that has substantial practical implications for prenatal breastfeeding education programs and the shaping of healthcare policy and practice.

Nevertheless, while adept at capturing complex perspectives, this descriptive qualitative design inherently limits generalizability due to its focus on depth rather than breadth, thus reflecting experiences specific to the 30 participants involved rather than a broader population. Additionally, online semi-structured interviews, although increasing accessibility, may have influenced participants' comfort and potentially affected the openness of responses compared to in-person interactions.

Furthermore, the reliance on self-reported data—gathered through Qualtrics questionnaires and scales, such as the Iowa Infant Feeding Scale for mothers and the HSE Self-Assessment Competency Framework for healthcare professionals—introduces the possibility of response bias, as participants may have provided responses they deemed favourable or socially acceptable.

Regarding data analysis, reflexive thematic analysis enabled the study to capture nuanced insights; however, the subjective nature of coding and theme development may reflect researcher biases despite efforts toward reflexivity throughout the analytical process. Despite these limitations, this study represents a meaningful contribution to understanding prenatal breastfeeding education, providing a foundation for future research and programmatic enhancement.

## Conclusion

This qualitative descriptive study reveals that while prenatal breastfeeding education classes in Ireland provide essential theoretical knowledge, they often lack the practical skills and emotional support that are necessary for successful breastfeeding. The findings point to a disconnect between the objectives of these classes and the actual experiences of postnatal mothers, who frequently feel unprepared and overwhelmed. A holistic approach to breastfeeding education is recommended to bridge this gap, integrating interactive, personalized, and culturally sensitive elements that better align with mothers' needs and expectations. Enhancing emotional support, providing standardized training for healthcare providers, and implementing hybrid learning models may ultimately improve breastfeeding outcomes and experiences for mothers.

## Supporting information

**S1 Table. COREQ checklist.**
(DOCX)

**S2 Table. GRIPP2 short form (PPI = patient and public involvement).**
(DOCX)

**S3 Table. Interview topic guides.**
(DOCX)

**S4 Table. Iowa Infant Feeding Attitude Scale (IIFAS).**
(DOCX)

**S5 Table. HSE Self-Assessment Competency Framework for breastfeeding support.**
(DOCX)

**S6 Table. Illustrative quotations presented by theme and sub-themes.**
(DOCX)

## Author Contributions

**Conceptualization:** Jennifer Kehinde, Claire O'Donnell, Annmarie Grealish.

**Data curation:** Jennifer Kehinde, Annmarie Grealish.

**Formal analysis:** Jennifer Kehinde, Claire O'Donnell, Annmarie Grealish.

**Investigation:** Jennifer Kehinde.

**Methodology:** Jennifer Kehinde, Claire O'Donnell, Annmarie Grealish.

**Project administration:** Jennifer Kehinde.

**Supervision:** Claire O'Donnell, Annmarie Grealish.

**Validation:** Annmarie Grealish.

**Visualization:** Claire O'Donnell, Annmarie Grealish.

**Writing – original draft:** Jennifer Kehinde, Claire O'Donnell, Annmarie Grealish.

**Writing – review & editing:** Jennifer Kehinde, Claire O'Donnell, Annmarie Grealish.

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
