## [Decision Letter · Decision Letter 0]

23 Oct 2024

PONE-D-24-33587The Perspectives of Prenatal Breastfeeding Educational Classes in Ireland and Their Influence on The Mother’s Decision to Breastfeed: A Qualitative StudyPLOS ONE

Dear Dr. Grealish,

Thank you for submitting your manuscript to PLOS ONE. After careful consideration, we feel that it has merit but does not fully meet PLOS ONE’s publication criteria as it currently stands. Therefore, we invite you to submit a revised version of the manuscript that addresses the points raised during the review process.

Thank you for your submission. To improve the reporting of your qualitative study, please follow the **COREQ 32-item checklist** for interviews and focus groups. Kindly revise the manuscript accordingly and attach the completed checklist with your resubmission.

We look forward to receiving your revised manuscript.

Kind regards,

Trhas Tadesse Berhe, PhD

Academic Editor

PLOS ONE

Journal Requirements: When submitting your revision, we need you to address these additional requirements. 1. Please ensure that your manuscript meets PLOS ONE's style requirements, including those for file naming. The PLOS ONE style templates can be found at https://journals.plos.org/plosone/s/file?id=wjVg/PLOSOne_formatting_sample_main_body.pdf and https://journals.plos.org/plosone/s/file?id=ba62/PLOSOne_formatting_sample_title_authors_affiliations.pdf 2. Please include your full ethics statement in the ‘Methods’ section of your manuscript file. In your statement, please include the full name of the IRB or ethics committee who approved or waived your study, as well as whether or not you obtained informed written or verbal consent. If consent was waived for your study, please include this information in your statement as well. 3. Please include a separate caption for each figure in your manuscript. 4. Please include captions for your Supporting Information files at the end of your manuscript, and update any in-text citations to match accordingly. Please see our Supporting Information guidelines for more information: http://journals.plos.org/plosone/s/supporting-information.

Reviewers' comments:

Reviewer's Responses to Questions

**Comments to the Author**

1. Is the manuscript technically sound, and do the data support the conclusions?

Reviewer #1: Yes

Reviewer #2: Yes

Reviewer #3: Partly

2. Has the statistical analysis been performed appropriately and rigorously? 

Reviewer #1: N/A

Reviewer #2: Yes

Reviewer #3: N/A

3. Have the authors made all data underlying the findings in their manuscript fully available?

Reviewer #1: Yes

Reviewer #2: Yes

Reviewer #3: No

4. Is the manuscript presented in an intelligible fashion and written in standard English?

Reviewer #1: Yes

Reviewer #2: Yes

Reviewer #3: No

5. Review Comments to the Author

Reviewer #1: The manuscript is interesting, comprehensive and well written.

Authors have followed the COREQ guidelines. However, authors need to explain why they did not try to comply with saturation of data? Is there any comment?

Other than that, the exclusion criterias were not the antonym of the inclusion criteria.

The length of the manuscript should be considered

Reviewer #2: Well done. Although this could benefit from a substantial cut to reduce word count, as it is thorough, the qualitative methods and analysis employed were rigorous and adequately described. I have suggested a few minor revisions.

Reviewer #3: Dear Author

The presented study is commendable and valuable. However, I have significant concerns. I hope the suggestions will help improve the quality of the article.

My comments on the manuscript are as follows:

The title needs modification.

I suggest removing the phrase "and their influence on the mothers decision to breastfeed".

This statement is quantitative in nature and is not suitable for the type of study, and on the other hand, according to the nature of the findings, it is not possible to achieve this goal.

Abstract

- Findings: In addition to the findings, conclusions from the findings are also presented in this section. This writing style has created ambiguity. I suggest that it be modified in terms of writing.

- Discussion:

The discussion in the abstract is not based on the findings of this study. It is suggested to modify the discussion based on the results.

Introduction:

- The phrase "This phrase should be removed from the target" should be removed from the target

Method:

-The exclusion criteria needs to be modified. Note that the withdrawal criterion includes cases where the participants are excluded from the study after entering the study.Please revise.

-Results:

-Data analysis should be modified. Illustrative codes are not properly compared and merged, or I cannot understand their semantic relationship with sub-themes and themes. Please use clearer labels. Or review the categories. For example, what are the illustrative codes for the sub-theme "Breastfeeding Confidence vs. Societal Norms"?

-How could you extract the theme "Quality of Prenatal Breastfeeding Education" with one sub-them "Consistency and quality of information received".

suggest you review the findings carefully and fix this basic problem in data analysis.

-In many of the explanations you have given for themes and sub-themes with quotations, you have used a quantitative approach that firmly describes the relationships between cause and effect. This is a major problem and needs modification. For example, in the phrase "postnatal mothers highlighted a gap between the educational content and the "holistic" preparation

needed for a successful breastfeeding experience. Some of the postnatal mothers felt that

"inadequate knowledge" about common breastfeeding challenges contributed to "unmet

expectations", negatively impacted their breastfeeding self-efficacy and subsequently affected

their decision to continue breastfeeding".

Is it possible to explain this causal relationship decisively based on the experience of one or two participants? Please pay attention to the explanations provided for themes and sub-themes and modify similar items.

Discussion:

-limitations

Weakness in generalizability is part of the nature of qualitative studies. For the limitations of your study, I suggest that you mention the limitations you had in conducting the study, including methodology and data analysis. Including the method of data collection and so on.

-

6. PLOS authors have the option to publish the peer review history of their article (what does this mean?). If published, this will include your full peer review and any attached files.

Reviewer #1: **Yes: **Yayi Suryo Prabandari

Reviewer #2: **Yes: **S. Alexandra Marshall

Reviewer #3: **Yes: **Mehrsadat Mahdizadeh

---

## [Decision Letter · Decision Letter 1]

25 Nov 2024

A Qualitative Study on the Perspectives of Prenatal Breastfeeding Educational Classes in Ireland: Implications for Maternal Breastfeeding Decisions

PONE-D-24-33587R1

Dear Dr. Annmarie Grealish,

We’re pleased to inform you that your manuscript has been judged scientifically suitable for publication and will be formally accepted for publication once it meets all outstanding technical requirements.

Kind regards,

Trhas Tadesse Berhe, PhD

Academic Editor

PLOS ONE

Additional Editor Comments (optional):

Reviewers' comments:

Reviewer's Responses to Questions

**Comments to the Author**

1. If the authors have adequately addressed your comments raised in a previous round of review and you feel that this manuscript is now acceptable for publication, you may indicate that here to bypass the “Comments to the Author” section, enter your conflict of interest statement in the “Confidential to Editor” section, and submit your "Accept" recommendation.

Reviewer #2: All comments have been addressed

Reviewer #3: All comments have been addressed

2. Is the manuscript technically sound, and do the data support the conclusions?

Reviewer #2: Yes

Reviewer #3: Yes

3. Has the statistical analysis been performed appropriately and rigorously? 

Reviewer #2: Yes

Reviewer #3: Yes

4. Have the authors made all data underlying the findings in their manuscript fully available?

Reviewer #2: Yes

Reviewer #3: Yes

5. Is the manuscript presented in an intelligible fashion and written in standard English?

Reviewer #2: Yes

Reviewer #3: Yes

6. Review Comments to the Author

Reviewer #2: I believe the authors have addressed all the comments provided by the reviewers. I have no additional comments to address.

Reviewer #3: Dear Author

Thank you for providing valuable findings to solve the problems of breastfeeding education classes and for your efforts to complete your manuscript.

7. PLOS authors have the option to publish the peer review history of their article (what does this mean?). If published, this will include your full peer review and any attached files.

Reviewer #2: **Yes: **S. Alexandra Marshall

Reviewer #3: **Yes: **Mehrsadat mahdizadeh

---

## [Editor Report · Acceptance letter]

3 Dec 2024

PONE-D-24-33587R1 

PLOS ONE

Dear Dr. Grealish, 

I'm pleased to inform you that your manuscript has been deemed suitable for publication in PLOS ONE. Congratulations! Your manuscript is now being handed over to our production team.

Kind regards, 

on behalf of

Dr. Trhas Tadesse Berhe 

Academic Editor

PLOS ONE